# The National ReferAll Database: An Open Dataset of Exercise Referral Schemes Across the UK

**DOI:** 10.3390/ijerph18094831

**Published:** 2021-04-30

**Authors:** James Steele, Matthew Wade, Robert J. Copeland, Stuart Stokes, Rachel Stokes, Steven Mann

**Affiliations:** 1Ukactive Research Institute, Ukactive, London WC1A 2SL, UK; matthewwade@ukactive.org.uk; 2Faculty of Sport, Health, and Social Sciences, Solent University, Southampton SO14 0YN, UK; 3The Advanced Wellbeing Research Centre, Sheffield Hallam University, Sheffield S9 3TU, UK; r.j.copeland@shu.ac.uk; 4The National Centre for Sport and Exercise Medicine, Sheffield S9 3TY, UK; 5ReferAll Ltd., Worthing BN11 1LY, UK; stuart@refer-all.net (S.S.); rachel.stokes@refer-all.net (R.S.); 64Global Consulting Ltd., London W4 5YG, UK; steven.mann@4global.com

**Keywords:** health database, exercise referral, physical activity, big data

## Abstract

In 2014, The National Institute for Health and Care Excellence (NICE) called for the development of a system to collate local data on exercise referral schemes (ERS). This database would be used to facilitate continued evaluation of ERS. The use of health databases can spur scientific investigation and the generation of evidence regarding healthcare practice. NICE’s recommendation has not yet been met by public health bodies. Through collaboration between ukactive, ReferAll, a specialist in software solutions for exercise referral, and the National Centre for Sport and Exercise Medicine, which has its research hub at the Advanced Wellbeing Research Centre, in Sheffield, data has been collated from multiple UK-based ERS to generate one of the largest databases of its kind. This database moves the research community towards meeting NICEs recommendation. This paper describes the formation and open sharing of The National ReferAll Database, data-cleaning processes, and its structure, including outcome measures. Collating data from 123 ERSs on 39,283 individuals, a database has been created containing both scheme and referral level characteristics in addition to outcome measures over time. The National ReferAll Database is openly available for researchers to interrogate. The National ReferAll Database represents a potentially valuable resource for the wider research community, as well as policy makers and practitioners in this area, which will facilitate a better understanding of ERS and other physical-activity-related social prescribing pathways to help inform public health policy and practice.

## 1. Background

The National Institute for Health and Care Excellence (NICE) published guidelines regarding exercise referral schemes (ERS) in 2014 [1]. The extant literature at the time regarding the impact of ERS was considered inadequate, with inconsistent and weak evidence regarding their effects upon health, wellbeing, and quality-of-life outcomes [1,2,3,4]. ERS can result in increased physical activity [5], but the impact of these programmes should also be assessed against broader health and wellbeing outcomes [6]. 

The world of ERS has been described as ‘wild and woolly’, lacking clarity in the conceptualisation of ‘exercise’ with minimal stakeholder agreement in how to determine ‘impact’ [7]. Henderson et al. [7] have argued that ERS do not work per se, but that their effectiveness is determined by the interpretations of their participants and whether they ‘improve’ on an individual basis. While important to the person, change at the individual level does not facilitate understanding of whether a real effect exists within a population, or the size of that effect and the precision with which it can be estimated. These are important factors to consider when developing policies and commissioning programmes and interventions. Indeed, it has been argued [8] that sport and exercise medicine has, for some time, been drowning in a body of evidence regarding ‘efficacy’ (the extent to which an intervention has the ability to bring about an intended effect under ideal circumstances, e.g., in laboratory settings) whilst simultaneously dying of thirst from a lack of evidence regarding ‘effectiveness’ (the extent to which an intervention achieves its intended effect in its usual setting). If improving population health is a goal of stakeholders, particularly those determining strategy and policy or delivering programmes and interventions, then research examining effects from ecologically valid datasets are needed to determine effectiveness of the existing delivery and compliment the evidence base examining efficacy. 

One of the recommendations made in the 2014 guidelines from NICE was that *“Public Health England should develop and manage a system to collate local data on exercise referral schemes”* and that these data should be based upon the Standard Evaluation Framework for physical activity interventions [1]. Furthermore, those data should be made available (in a useable format) for analysis and research to inform future practice. ‘Big data’ analytics is a current trend in healthcare, and it has been argued that within it lies the potential to transform the way policy, commissioning, and delivery decisions are made in healthcare contexts [9]. Indeed, the use of health databases has been argued to have a considerable impact upon promotion of scientific endeavour [10]. Excellent work regarding the large Welsh National Exercise Referral Scheme has been conducted linking with routine health records and reported in this special issue [11]. However, an open resource such as this for ERS across the UK with a focus on outcome data has yet to be produced. 

The National ReferAll Database is a newly formed resource produced by ukactive, the National Centre for Sport and Exercise Medicine in Sheffield, and ReferAll, which includes data on a variety of outcomes for patients participating in ERS. The National ReferAll Database represents the largest open database, and in essence is the largest longitudinal study, of ERS in the UK. As a result of the initial work to develop The National ReferAll Database, NICE, following consultation, updated the guidance on ERS with the need for a national database removed [12]. The potential of the National ReferAll Database, we believe, is significant. It offers researchers, policy makers, practitioners, and clinician’s access to myriad observational data through which they can explore what might work, and what might not, in terms of ERS. 

In 2019, we described the initial formation of a phase 1 data cut which was subsequently used by our research groups for several studies presenting initial insights from the existing data [13,14,15]. We are now in a position with a phase 2 data cut and appropriate permissions to make it openly available to the research community with an interest in ERS. The aim of this paper is to describe the formation and open sharing of The National ReferAll Database including the data cleaning processes, structure, which outcome measures are included, and an explanation of the intended use case. Further, limitations of the database alongside future ambition for development are discussed. Lastly, we call upon researchers, practitioners, and policy makers to engage with The National ReferAll Database to facilitate evaluation of ERS and other physical-activity-related referral schemes in the future.

## 2. Materials and Methods

The present cut represents data from between 2012 and March 2021 and includes 39,283 unique persons having been referred for an ERS. Participants had been referred to one of 123 ERSs across the UK. Data are captured by the referring organisation regarding the referral, which is recorded in ReferAll’s database. Then, at the level of the ERS, additional data are captured regarding the referral’s participation in the ERS, in addition to referral outcome data. In essence, this database represents a longitudinal cohort observation study design following individuals entering an ERS. 

### 2.1. Global Data Protection Regulation

All data were handled in accordance with the Global Data Protection Regulations (GDPR). Only anonymized data were ever handled by the research team, who prepared the dataset for open sharing. All data were shared with informed consent. Considering the retrospective nature of the study design evaluating existing data, and that no identifiable data were involved, all data were handled in accordance with GDPR, and guidance from the Health Research Authority and Research Ethics Committee Section 11 of Standard Operating Procedures regarding Health Databases, a priori ethical/IRB approval was not required for this work.

### 2.2. Data Preparation and Cleaning

All data processing and cleaning was conducted in R (v 4.0.2; R Core Team, https://www.r-project.org/, accessed on 21 April 2021). For participating schemes, a data export was prepared containing each unique referral and their scheme characteristics, referral pathways, and demographic data. Additional exports for each participating scheme were produced containing all outcome data fields and responses. Initial data exports contained postcode data for participants for which Index of Multiple Deprivation, Lower Layer Super Output Area, and Rural Urban Classifications were determined. Postcodes were then removed from the data for the purpose of open sharing to reduce the possibility of re-identification. Considering recent work modelling the ability to re-identify anonymised data from demographic attributes, we have attempted to limit them in this dataset as much as possible [16].

Outcome data exports highlighted the considerable variety of outcomes, primarily through ad hoc questionnaire designs employed by schemes. Across the participating schemes, there were 1054 unique outcome fields (note, that this includes separate questions within questionnaires as unique). Thus, we manually screened these to identify and extract outcome measures from known questionnaire tools, or physical measures. The outcomes included in the dataset are described separately below. We would be happy to work with additional researchers wanting to review these fields to identify other outcomes of interest to include in the dataset for future versions.

Data cleaning [17] was performed due to the size of the database and the fact that data were input manually at the source of collection by ERS staff. As noted, the data cut used here contained data from 39,283 participants; however, an examination of the data highlighted that most outcomes contained data outside of plausible ranges based upon the measure and its unit of measurement. It is likely that this was the result of data entry errors. Thus, values were filtered with respect to their range and upper and lower cut-offs used to exclude data including possible ranges for physical variables, and possible ranges for questionnaire-based data considering scoring systems used. Any questionnaire with existing guidance for cleaning and processing were tackled following these guidelines. Table 1 shows the cut-offs used for data exclusion. Data editing was not performed (the exception being for questionnaires wherein the standard guidelines required this) as, in most cases, it was impossible to determine satisfactory rules to address potential reasons for incorrect input of values (e.g., in the height field, a value may read ‘69’ and, as this field is supposed to be in cm, it is plausible that the person entering the data accidentally missed the ‘1’ of the start and it should be ‘169 cm’, in addition to its being plausible that they incorrectly entered it in inches, which would mean it should be ‘175.3 cm’). 

## 3. Description of the Dataset

A data dictionary is available at the Open Science Framework page where The National ReferAll Database is hosted (see section below). This contains the column number references, variable names, variable descriptions, variable types, and options or ranges. The full dataset is also available through the project page, though we encourage users to first view the dictionary and plan questions/hypotheses and analysis plans before diving into the dataset proper. Below, we present for the reader characteristics of the current dataset, though note that future readers may find that additional and updated versions are now available on the project page.

### 3.1. Scheme and Referral Characteristics

A range of scheme and referral level characteristics are included in the dataset. These are primarily those captured by the referring organisations for input to ReferAll. These are presented in Table 2 with descriptive characteristics. 

We have also included three open field string variables in the dataset: reason for referral, reasons for not participating, and reason for leaving early. These open fields are completed by ERS staff and, as such, there is little standardisation of how data is input. However, given advances in use of natural language, text mining with healthcare records [27] and the proliferation of free guides, software, and packages to support this (e.g., [28]), we felt it was appropriate to include this potentially rich source of information for researchers to access.

### 3.2. Outcome Measures

As noted, outcome measures captured varied across the ERS. Physical activity was commonly assessed using the International Physical Activity Questionnaire (IPAQ) short form [18]. In addition to this, both the Single Item Metric and Short Active Lives tools [19] from Sport England for measuring self-reported physical activity were captured by some schemes. Quality of life was captured by schemes using either the EQ-5D-5L [20] (this was processed using the crosswalk conversion value set for the UK to produce health-related quality-of-life scores) or the World Health Organization Well-Being Index (WHO-5) [21]. Mental health and wellbeing were captured using the full and short versions of the Warwick–Edinburgh Mental Wellbeing Scale Warwick Edinburgh Mental Wellbeing Scale (WEMWBS/SWEMWBS) [22]. More specific to ERS, Exercise Related Quality of Life scale (ERQoL) [24] and the Exercise Self-Efficacy Scale [23] were captured by some schemes. Physical measures included: height, weight, body mass index, waist and hip circumference, waist:hip ratio, systolic blood pressure (SBP), diastolic blood pressure (DBP), resting heart rate, and 30 s sit-to-stand scores. These are presented in Table 3 with descriptive characteristics.

### 3.3. Open Science Framework Hosting

To facilitate the sharing of this dataset, we opted to host it on the freely accessible Open Science Framework (https://osf.io, accessed on 21 April 2021) platform operated by the Centre for Open Science (https://www.cos.io/, accessed on 21 April 2021). The platform offers a variety of tools and services to assist in open research practices [29], and thus we felt it was appropriate for hosting The National ReferAll Database. Through the Open Science Framework hosting platform, we can provide data dictionaries detailing the content of the datasets, and direct access to these datasets by users. Note that datasets are version controlled by date (YYYY-MM-DD format), and so future readers may note that the platform contains multiple versions. We also intend to collate and upload reports from users of the dataset. With respect to the datasets shared, these are under Creative Commons Attribution 4.0 International Public License. The database is available at https://osf.io/uzbw9/, accessed on 21 April 2021 and Figure 1 shows the landing page for the project. 

Users can access data without restriction, though we would encourage users to consider the following and to utilize the functionality of the Open Science Framework platform:Create a dedicated project page to manage research you intend to conduct using The National ReferAll Database datasets. Here, you should host any wider materials, additional data (or note with attribution the relevant version of The National ReferAll Database datasets being used), and any analysis code used;Develop proposed research questions or hypotheses to test after initially considering the data dictionaries and then prepare appropriate analysis plans to publicly pre-register on this project page. If not pre-registering, ensure that you report your analysis as purely exploratory;Report the results of any analysis first by pre-prints with an open licence through an appropriate pre-print server (e.g., https://osf.io/preprints/sportrxiv/, accessed on 21 April 2021) including links to any subsequent publications in other outlets (i.e., peer-reviewed journal articles).

We feel that following simple practices such as these will help to serve the principles of openness, transparency, and reproducibility upon which The National ReferAll Database has been formed.

## 4. Discussion

Despite widespread adoption, research exploring the effect of ERS on health outcomes from ecologically valid datasets is scarce. This manuscript describes the formation of The National ReferAll Database; an open UK-wide database of outcome data from participation in ERS. Continued evaluation will improve both the delivery and effectiveness of ERS and other physical-activity-related referral schemes. Indeed, the formation and continued development of The National ReferAll Database presents several opportunities to further our understanding of ERS.

‘Big data’ analytics has potential to transform the way in which policy and commissioning decisions are made across health and care [9]. The National ReferAll Database in theory meets the definition of ‘big data’ (defined as: *Log* [*n* * *p*] ≥ 7; [30]) Considering the number of observations rows (*n* = 45,181) and variable columns (*p* = 52), The National ReferAll Database has a total cell count of 1,155,099 and a *Log* [45,181 * 52] of 14.67. As more ERS join the database, the National ReferAll Database will truly lend itself to ‘big data’ analytics, enabling policy makers, commissioners, researchers, and practitioners to make more precise decisions as to which interventions might work best for whom and under what circumstances. 

Findings from analysis of the phase 1 data cut suggest that, broadly speaking, the effects of ERS on population level physical activity are limited, and there is also minimal effect on health and wellbeing outcomes [13,15]. These somewhat disappointing insights are likely skewed by heterogeneity in the scheme design, delivery and outcome measures adopted. To improve our understanding of ERS, there is a need for consistency in outcome measures and this could be driven via the National ReferAll Database. With respect to exercise-based interventions generally, there is less doubt as to the efficacy of exercise for improving various health and wellbeing related outcomes. Indeed, a goal for future iterations of The National Referral Database is to enable data linkage with electronic health records to consider outcomes such as healthcare use. In their current form, however, evidence suggests that ERS do not deliver the intervention effects that might be expected based upon the apparent efficacy of exercise as a therapy. The continued growth and use of The National ReferAll Database will enhance the standardisation of ERS with respect to intervention components and treatment fidelity, and this is likely to positively influence the quality of ERS and their effect on health-related outcomes. Standardisation is challenging for a host of sociological reasons, including resources, competencies, and levels of engagement [7]. To address these barriers to standardisation, Henderson et al. [7] suggests that an obvious place to start is with a robust evidence base, on which there is at least agreement on what does work best at a population level. However, understanding this requires greater homogeneity in reporting to support evaluation. Hanson et al. [31] have recently developed, through a Delphi process, a taxonomy for reporting and classifying physical activity referral schemes including ERS. Through following such an approach, the implementation of standardisation then, at least, has a starting point, and indeed many, though not all, aspects of this are captured in The National Referral Database at present. 

The apparent lack of effectiveness of ERS based on present evidence could also be explained through issues with the translation and implementation of interventions used in standardized settings (such as with supervised ERS employed in RCTs) in reflecting local needs. Indeed, intervention fidelity is an issue of considerable importance for healthcare interventions [32]. Considering this, a new framework for the co-production of ERS alongside multidisciplinary stakeholders as a novel approach has been developed [33]. Further, though evidence at present is limited [34], the use of wider social prescribing to address other areas of support that individuals might need to tackle first, to facilitate their wider health and wellbeing, is widely promoted. This includes the recent National Health Service (NHS) *Long Term Plan* [35]. Link worker social prescribing has been suggested as an approach to help achieve this, whereby support is provided to patients across wider personal issues [36]. It seems likely that there is a need for both standardisation to ensure that ERS and other physical activity referral schemes are evidence-based with respect to their ability to produce the desired effects, in addition to co-production of ERS to account for the nuances of local context. This might ensure that implementation considers important personal and social barriers to ensuring ERS can succeed in achieving this. 

Another current trend that runs parallel to that of ‘big data’ is ‘precision medicine’, albeit this is not without criticism [37]. Initial analysis from phase 1 evidenced considerable heterogeneity between ERSs for all outcomes [13,15] but, importantly, also for the demographics, conditions, and pathways of the participants. This suggests that individual-level factors could explain the responsiveness of outcomes to ERS. With this in mind, there may be a benefit to identifying who responds best to which interventions and what the predictors are at the individual level. Indeed, ‘big data’ has been argued to be valuable in the quest towards ‘precision medicine’ [38] and the establishment of The National ReferAll Database could, in theory, contribute to that end as it develops. The data included are pre-, post-, and, in some cases, further follow-ups, for outcomes. However, without the use of appropriate controls from which to determine whether *true* inter-individual variation is indeed identifiable for these outcomes, it is difficult to identify meaningful predictors [39]. Perhaps a better understanding of what specific populations (e.g., those referred with type II diabetes, or cardiovascular disease, or musculoskeletal disorders [5]), and which specific types of ERS work best comparatively, or indeed a combination of these two (i.e., what works best for which population), may be the next step in the generation of evidence to help guide the implementation of ERSs. In addition to understanding, at a population level, what effects ERS may have upon those with specific conditions (indeed, this is often the most reliable indicator of individual level effects [39]), realist reviews of what approaches to ERS work, for whom, and within what context will add value to the translation of observed effects to different populations and patient groups. This approach has already been adopted to understand wider social prescribing schemes [40,41].

The National ReferAll Database, along with the phase 1 analyses previously presented, are not without several limitations. Briefly, the data shown here are observational in nature (in essence, they show the change in participants of ERSs over time). Thus, similarly to the determination of inter-individual variation in outcomes, without an appropriate control group, any changes in outcomes lack a counterfactual and appropriate caution must be given to their meaning. In prior analyses, and where possible, we based our tests upon null intervals (as opposed to point nulls, i.e., a change of zero) around what were considered as minimal clinically important changes in the extant literature. However, it could be that these are inappropriate, and indeed many of the statistically significant changes could also be deemed meaningful. However, other evidence from RCTs suggests that, for many of the outcomes examined, exercise can produce statistically significant and meaningful changes (see discussions in [13,15]), but that, in the ecologically valid examination of ERS, these changes do not manifest in the data to the same degree—possibly due to other confounding variables. 

Details of the schemes, classifications and characteristics included in the database are incomplete, something which our phase 1 analyses were unable to consider. Though many elements of the newly developed taxonomy from Hanson et al. [31] are captured in the database, and in this phase 2 data cut, we were able to include the type of scheme, many other key characteristics are not captured. For example, the database currently lacks information regarding the prescription of exercise provided in each scheme using consistent criteria (e.g., FITT) This is a major limitation in drawing conclusions regarding what might work in terms of specific ERS. It also undermines the strategic implementation of schemes, whether private or public sector, as there appears to be differences in how ‘exercise’ or ‘physical activity’ is conceptualized [7]. Evidence from cardiac rehabilitation programmes suggests that many participants do not meet the intended intensity of effort for exercise prescription [42,43]. This could help explain the lackluster results observed here for ERS. Further, psychosocial and behaviour change techniques which might influence uptake, attendance, and adherence are rarely reported [44,45]. We have recently conducted a survey study of ERS schemes utilizing the Consensus on Exercise Reporting Template (CERT; [46]) which supports these concerns [47]. The collection of descriptive data regarding the ‘active components’ of ERS is something which we are working towards for The National ReferAll Database in future iterations. Further, monitoring of intervention fidelity utilizing known treatment fidelity frameworks will be essential to enhance the understanding of the feasibility of wider implementation should effectiveness be identified [32].

Our phase 1 analyses did not consider participant characteristics (e.g., age, sex, disability, employment status, ethnicity, socioeconomic status, referral pathways, referral reasons etc.), or how these might moderate changes over time from participation. This is partly due to the variety of present options with respect to many of these variables are currently captured. However, similar issues apply here with respect to determining the impact of these factors for the individual in relation to predicting responsiveness without sufficient understanding of typical long-term variability in outcome measures independently of any intervention (i.e., to identify if someone is a *true* high or low individual responder [39]). The inclusion of monitored wait-list controls for any schemes that use them in the database could improve research designs, allowing for the use of quasi-experimental interrupted time-series analyses, and address, to some degree, the problem of the counterfactual, thus improving the ability to make causal inferences. Such an inclusion might permit the National ReferAll Database to facilitate large simple trials in a similar manner to other databases and electronic health records [48]. Future work is already planned and/or underway to explore drop out/loss to follow up, the outcomes routinely collected from ERS and their appropriateness, in addition to whether ERS schemes are presently targeting and including those for whom they are intended based upon NICE guidelines. 

Missing data is also currently a limitation with any ecological health data, and one which is also a concern for The National Referral Database. As noted, the database in theory meets the definition of ‘big data’ (defined as: *Log* [*n* * *p*] ≥ 7; [30]), with a *Log* value of 14.67. However, when we consider that missing data represents 51% of all cells in the dataset, this reduces to a *Log* value of 7.26, meaning it only just passes this threshold. Missing data in health records can represent a challenge for modelling, whereby the imputation or removal of incomplete records can bias performance [49], although, in some cases missing data can be considered informative where the absence of data might represent information on some underlying variable [50]. Nevertheless, users of the dataset should be cautious in their inferences regarding models for this reason. We have ambitions to address this in future iterations.

Lastly, representativeness of the dataset should be considered. This open phase 2 dataset is smaller (*n* = 39,283) in terms of the number of unique referrals than might be expected given the number of ERS included in comparison to the phase 1 dataset used (*n* = 23,782). The reason for this is that we did not have the requisite consent in place in line with GDPR for the open sharing of that phase 1 dataset through the Open Science Framework. As such, we had to go back out to schemes to capture the requisite consents. Unfortunately, this occurred during 2020, when the COVID-19 pandemic resulted in many schemes stopping operations and staff being put on furlough. Due to this, we were not able to obtain consent from some large schemes which had originally consented to sharing data in phase 1. In addition to the logistical difficulties in obtaining consent, we should consider the possible role of self-selection bias. Schemes who believe themselves to be effective might have been more likely to consent to data sharing, whereas those who did not believe themselves to be as effective might have shied away from this. We hope that a culture shift towards open data and continued evaluation and improvement might help encourage more schemes to share data to improve the representativeness of the dataset in future iterations. 

## 5. Conclusions

The National ReferAll Database represents a valuable resource for the wider research community, as well as policy makers and practitioners in physical activity and public health. It has the potential to facilitate a better understanding of ERS and other physical activity referral pathways to help inform public health policy and practice. With the establishment of The National ReferAll Database as an open resource, we have ambitions to continually update it with additional data and version controls, for researchers to access and policy makers and practitioners to use to inform their policies/practices. 

In addition to the ambitions noted in the discussion to improve the quality of The National ReferAll Database (capturing further details of ERS including the specific exercise components, fidelity of interventions, inclusion of wait-list controls, evaluation of captured outcomes etc.), it is hoped that the database will grow to incorporate new pathways and schemes outside of the traditional conceptualization of the ERS. Indeed, with the COVID-19 pandemic, many schemes have had to adapt delivery, although, even prior to this, the use of online or virtual delivery of ERSs has been explored (e.g., [51]). The inclusion and classification of different schemes will enable their comparisons. 

It is hoped that insights from the National ReferAll Database will be incorporated into practice and policies locally, nationally, and globally and, thus, as data are fed back into the database in future iterations from schemes as they adapt, this will facilitate continued evaluation. As the introduction of novel schemes and approaches occurs, their evaluation in comparison to established standards can be facilitated. In considering the widening interest in social prescribing [35,52], linking with established networks of alterative referral pathways that fit within the realm of physical activity may also foster innovation in this space. We anticipate being able to report on these advancements in future papers in keeping with the acknowledgment from NICE regarding our efforts [12].

We have liaised directly with the wider community that we anticipate will utilise The National ReferAll Database, including:Academics and researchers who have expressed interest in the existing data, in addition to plans to enhance the database, and what additional data might be needed to answer key research questions relating to this area;Policy makers and stakeholders from key organisations who have expressed interest in the resource and findings generated from it to inform strategy, policy, and investment in this area;Practitioners who work within this area who have also expressed interest in the findings generated in order to help guide their practice.

We conclude with a call to the wider community with an interest in ERS, that we have not yet reached. The success of this endeavor relies upon the demand for the potential insight that a resource such as The National ReferAll Database might provide. Those who are delivering ERS or other physical-activity-related schemes are encouraged to capture data and feed it into the database once the platform has been developed, researchers are welcome to engage with it to answer key research questions regarding these schemes, and policy makers and practitioners are free to utilize these insights to inform what they do.

## Figures and Tables

**Figure 1 ijerph-18-04831-f001:**
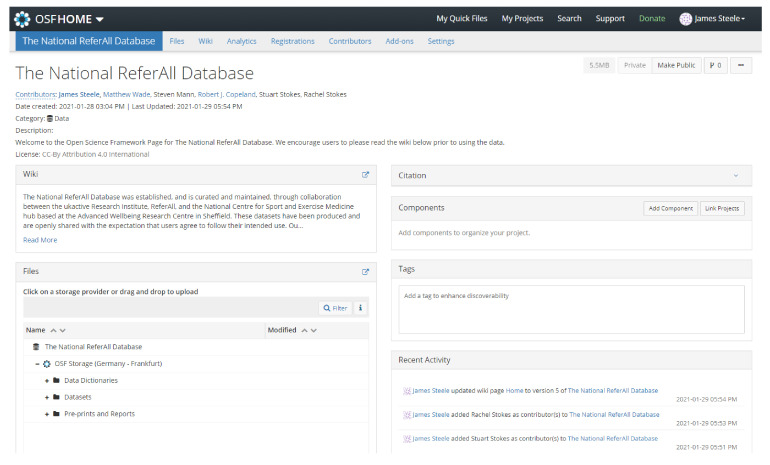
View of the Open Science Framework page for The National ReferAll Database.

**Table 1 ijerph-18-04831-t001:** Range of values for data cleaning.

Variable	Measurement and Units	Range	Source
International Physical Activity Questionnaire—Short Form	MET/Minutes and Categorical	Standard cleaning and analysis procedures for the IPAQ were followed	[18]
Short Active Lives	Days Per Week and Minutes Per Week and Categorical	Standard cleaning and analysis procedures for the Short Active Lives were followed	[19]
EQ-5D-5L	Visual Analogue Scale (0–100%) and Health Related Quality of Life (0–1)	Standard cleaning and analysis procedures for the EQ-5D-5L were followed using the crosswalk conversion value set for the UK	[20]
World Health Organization Well-Being Index	Total Score (0–25 pts)	Cleaned to range 0 pts to 25 pts i.e., impossible scores based on scale were removed	[21]
Warwick–Edinburgh Mental Wellbeing Scale	Total Score (0–70 pts)	Cleaned to range 14 pts to 70 pts i.e., impossible scores based on scale were removed	[22]
Short Warwick–Edinburgh Mental Wellbeing Scale	Total Score (0–35 pts)	Cleaned to range 7 pts to 35 pts i.e., impossible scores based on scale were removed	[22]
Exercise Self Efficacy Scale	Total Score (4–40 pts)	Cleaned to range 4 pts to 40 pts i.e., impossible scores based on scale were removed	[23]
Exercise Related Quality of Life Scale	Total Score (22–110 pts)	Cleaned to range 22 pts to 110 pts i.e., impossible scores based on scale were removed	[24]
Height	Centimetres (cm)	Cleaned to range of 122.5 cm to 205 cm	[25]
Weight	Kilograms (kg)	Cleaned to range of 40 kg to 180 kg	[25]
Body Mass Index	Kilograms Per Metre Squared (kg.m^2^)	Cleaned to range of 12 kg.m^2^ to 75 kg.m^2^	[25]
Waist Circumference	Centimetres (cm)	Cleaned to range of 20 cm to 197 cm	[25]
Hip Circumference	Centimetres (cm)	Cleaned to range of 30 cm to 195 cm	[25]
Resting Heart Rate	Beats Per Minute (*f*_c_)	Pre/post cleaned range to 40 *f*_c_ to 110 *f*_c_	[26]

**Table 2 ijerph-18-04831-t002:** Scheme and Referral Characteristics.

Characteristic	N = 39,283 ^1^
*Scheme Type*	
Cancer Rehab	74 (0.2%)
Cardiac Rehab	416 (1.1%)
ESCAPE Pain	526 (1.3%)
Exercise on Referral	30,590 (78%)
Falls Prevention	2366 (6.0%)
Physical Activity	89 (0.2%)
Pulmonary Rehab	162 (0.4%)
Specialist Exercise on Referral	174 (0.4%)
Stroke Rehab	54 (0.1%)
Swim4Health	182 (0.5%)
Weight Management	4650 (12%)
*Scheme Length (days)*	
42	292 (0.7%)
54	417 (1.1%)
84	20,032 (51%)
90	12,019 (31%)
168	6224 (16%)
175	299 (0.8%)
42	292 (0.7%)
*Referrer Organisation Type*	
Community	3753 (9.6%)
Hospital	2580 (6.6%)
Housing	5 (<0.1%)
Medical Centre	6723 (17%)
Outreach	2262 (5.8%)
Pharmacy	70 (0.2%)
School	2 (<0.1%)
Surgery	23,888 (61%)
*Referrer Type*	
Adult Nurse	4276 (11%)
Advanced Nurse Practitioner	183 (0.5%)
Alcohol Liaison Nurse	2 (<0.1%)
Art Therapist	5 (<0.1%)
BB Nurse	3 (<0.1%)
Cancer Nurse Specialist	40 (0.1%)
Cardiac Nurse	226 (0.6%)
Cardiac Physiologist	35 (<0.1%)
Cardiac Physiotherapist	51 (0.1%)
Cardiac Rehab Professional	41 (0.1%)
Cardiac Sister	25 (<0.1%)
Cardiologist	4 (<0.1%)
Change Coach	2 (<0.1%)
Clinical Nurse Specialist	8 (<0.1%)
Clinical Psychologist	9 (<0.1%)
Community Diabetes Team	9 (<0.1%)
Community Dietitian	11 (<0.1%)
Community Mental Health Worker	102 (0.3%)
Community Midwife	13 (<0.1%)
Community Physiotherapist	75 (0.2%)
Community Psychiatric Nurse	35 (<0.1%)
Consultant	41 (0.1%)
Consultant Psychiatrist	3 (<0.1%)
Counsellor	64 (0.2%)
Critical Care Technologist	4 (<0.1%)
Dietitian	89 (0.2%)
Doctor	8674 (22%)
Drama Therapist	1 (<0.1%)
Exercise Specialist	226 (0.6%)
Family Support Worker	23 (<0.1%)
General Practitioner	7985 (20%)
Gynaecologist	6 (<0.1%)
Health Education and Promotion Officer	72 (0.2%)
Health Improvement Officer	256 (0.7%)
Health Improvement Practitioner	54 (0.1%)
Health Professional	5 (<0.1%)
Health Support Worker	1832 (4.7%)
Health Trainer	364 (0.9%)
Health Trainer Coordinator	40 (0.1%)
Health Visitor	14 (<0.1%)
Healthcare Assistant	804 (2.0%)
Healthy Lifestyle Motivator	63 (0.2%)
Key Worker	6 (<0.1%)
Lead Nurse Diabetes	1 (<0.1%)
Learning Disability Nurse	1 (<0.1%)
Mental Health Nurse	111 (0.3%)
Mental Health Practitioner	42 (0.1%)
Mental Health Support Worker	82 (0.2%)
Mental Health Worker	52 (0.1%)
Midwife	21 (<0.1%)
Neurosurgeon	1 (<0.1%)
NHS Health Check Nurse	4 (<0.1%)
Nurse	5077 (13%)
Nursing Assistant	2 (<0.1%)
Occupational Therapist	123 (0.3%)
Orthopaedic Technician	5 (<0.1%)
Other Health Professional	667 (1.7%)
Paediatrician	2 (<0.1%)
Paramedic	2 (<0.1%)
Pharmacist	126 (0.3%)
Phlebotomist	1 (<0.1%)
Physiotherapist	3281 (8.4%)
Physiotherapy Assistant	97 (0.2%)
Podiatrist/Chiropodist	10 (<0.1%)
Practice Nurse	2445 (6.2%)
Prevention Worker	17 (<0.1%)
Psychiatrist	4 (<0.1%)
Psychologist	26 (<0.1%)
Psychotherapist	158 (0.4%)
Pulmonary Physio	46 (0.1%)
Recovery Workers	532 (1.4%)
Respiratory Physiology Technician	1 (<0.1%)
Respiratory Therapist	52 (0.1%)
Rheumatology Nurse	5 (<0.1%)
Senior Health Trainer	192 (0.5%)
Senior Physiotherapist	142 (0.4%)
Social Prescriber	14 (<0.1%)
Social Worker	40 (0.1%)
Specialist Health Improvement Practitioner	23 (<0.1%)
Staff Nurse	30 (<0.1%)
Technical Instructor	16 (<0.1%)
Therapy Assistant	51 (0.1%)
*Referral Status*	
Completed	15,680 (40%)
Intends To Participate	518 (1.3%)
Left Early	8027 (20%)
Not Participating	9590 (24%)
Participating	3721 (9.5%)
Referred	1747 (4.4%)
*Index of Multiple Deprivation (percentile)*	26 (14, 35)
Unknown	19,250
*Quintile of Deprivation*	
1	6176 (19%)
2	5859 (18%)
3	6092 (18%)
4	6891 (21%)
5	8364 (25%)
Unknown	5901
*Rural and Urban Classification (RUC11)*	
Rural town and fringe	2199 (6.0%)
Rural town and fringe in a sparse setting	5 (<0.1%)
Rural village and dispersed	1445 (4.0%)
Rural village and dispersed in a sparse setting	10 (<0.1%)
Urban city and town	17,947 (49%)
Urban major conurbation	13,905 (38%)
Urban minor conurbation	948 (2.6%)
Unknown	2824
*Referral in Scheme Area?*	
Yes	38,627 (98%)
No	656 (2%)
*Ethnic Group*	
Asian	256 (0.7%)
Black	1272 (3.2%)
Mixed	203 (0.5%)
Other	149 (0.4%)
Unknown/Withheld	26,650 (68%)
White	10,753 (27%)
*Ethnicity*	
African	551 (1.4%)
Arab	17 (<0.1%)
Bangladeshi	20 (<0.1%)
British	10,328 (26%)
Caribbean	621 (1.6%)
Chinese	10 (<0.1%)
Gypsy or Irish Traveller	2 (<0.1%)
Indian	93 (0.2%)
Irish	81 (0.2%)
Other Asian background	95 (0.2%)
Other Black background	100 (0.3%)
Other Ethnic group	132 (0.3%)
Other Mixed background	68 (0.2%)
Other White background	342 (0.9%)
Pakistani	38 (<0.1%)
Unknown	3 (<0.1%)
Unknown/Withheld	26,647 (68%)
White and Asian	19 (<0.1%)
White and Black African	40 (0.1%)
White and Black Caribbean	76 (0.2%)
*Age at Referral (years)*	52 (39, 63)
*Gender*	
Female	26,384 (67%)
Male	12,795 (33%)
Transgender	30 (<0.1%)
Unknown	74 (0.2%)
*Employment Status*	
Carer	135 (0.3%)
Employed full time	429 (1.1%)
Employed full time/part time	380 (1.0%)
Employed part time	338 (0.9%)
Full-time Student	6 (<0.1%)
Full time carer	16 (<0.1%)
Intermediate	59 (0.2%)
Long term sick/disabled	18 (<0.1%)
Look after home of family	6 (<0.1%)
Looking after home/family full time	32 (<0.1%)
Managerial/Professional	102 (0.3%)
Other	142 (0.4%)
Permanently sick/disabled	134 (0.3%)
Retired	2893 (7.4%)
Routine & Manual	71 (0.2%)
Self-employed	73 (0.2%)
Sick/Disabled/Unable to Work	83 (0.2%)
Student	71 (0.2%)
Unemployed	1747 (4.5%)
Unknown	32,455 (83%)
Volunteer	9 (<0.1%)
Unknown (not completed)	84
*Marital Status*	
Civil partnership	3 (<0.1%)
Co-habiting	53 (0.1%)
Divorced	96 (0.2%)
Married	919 (2.3%)
Other	3 (<0.1%)
Prefer not to say	55 (0.1%)
Separated	15 (<0.1%)
Single	132 (0.3%)
Unknown	37,887 (96%)
Widowed	120 (0.3%)
*Sexual Orientation*	
Bi/Bisexual	18 (<0.1%)
Gay man	2 (<0.1%)
Gay woman/Lesbian	8 (<0.1%)
Gay/Lesbian	19 (<0.1%)
Heterosexual/Straight	2892 (7.4%)
Other	6 (<0.1%)
Prefer not to say	578 (1.5%)
Unknown	35,760 (91%)
*Number of Outcome Measures (times)*	
1	35,480 (90%)
2	2061 (5.2%)
3	1475 (3.8%)
4	185 (0.5%)
5	78 (0.2%)
6	4 (<0.1%)

^1^ n (%); Median (IQR).

**Table 3 ijerph-18-04831-t003:** Outcome Measures.

Characteristic	N = 39,283 ^1^
*IPAQ (MET/Minutes)*	396 (0, 1386)
Unknown	30,642
*IPAQ (Category)*	
1	5263 (61%)
2	2330 (27%)
3	1048 (12%)
Unknown	30,642
*Sport England Single Item Metric (days)*	
0	2079 (35%)
1	1060 (18%)
2	670 (11%)
3	578 (9.7%)
4	316 (5.3%)
5	373 (6.2%)
6	126 (2.1%)
7	787 (13%)
Unknown	33,294
*Short Active Lives (Total Minutes)*	32 (0, 176)
Unknown	38,855
*Short Active Lives (Category)*	
1	205 (48%)
2	103 (24%)
3	120 (28%)
Unknown	38,855
*EQ-5D-5L (Visual Analogue Scale)*	50 (50, 50)
Unknown	36,220
*EQ-5D-5L (Health Related Quality of Life)*	1.00 (0.84, 1.00)
Unknown	36,220
*WHO-5 (%)*	56 (36, 68)
Unknown	35,804
*WEMWBS (pts)*	49 (41, 56)
Unknown	38,532
*SWEMWBS (pts)*	26.0 (22.0, 30.0)
Unknown	35,973
*Weight (kg)*	86 (73, 100)
Unknown	33,675
*Height (cm)*	167 (161, 174)
Unknown	33,794
*Body Mass Index (kg.m^2^)*	30 (26, 35)
Unknown	33,799
*Waist Circumference (cm)*	102 (90, 114)
Unknown	38,548
*Hip Circumference (cm)*	111 (100, 120)
Unknown	38,984
*Waist:Hip Ratio*	0.93 (0.86, 1.00)
Unknown	38,985
*Systolic Blood Pressure (mmHg)*	130 (119, 143)
Unknown	35,681
*Diastolic Blood Pressure (mmHg)*	81 (74, 89)
Unknown	35,681
*Resting Heart Rate (f*_c_)	78 (69, 86)
Unknown	35,808
*30 Second Sit to Stand (n)*	4.0 (0.0, 10.0)
Unknown	38,993
*Exercise Self Efficacy Scale (pts)*	31.0 (28.0, 35.0)
Unknown	39,195
*Exercise Related Quality of Life (pts)*	74 (65, 82)
Unknown	37,528

^1^ Median (IQR); n (%).

## Data Availability

The dataset described is available at https://osf.io/uzbw9/, accessed on 21 April 2021.

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
