# Peer review of "The National ReferAll Database: An Open Dataset of Exercise Referral Schemes Across the UK"

_ijerph, 2021, doi:10.3390/ijerph18094831_

Round 1

Reviewer 1 Report

First of all, I would like to thank the authors for having carefully read the comments of the previous review and for having responded to them, providing justification as to why they have decided not to attend to the suggestions made, mainly due to the limitations derived from the data collected in the database or from the impossibility of analyzing them.

As I indicated above, I consider that the Project is of great interest, and that the paper is well structured and presented, but it needs further analysis of variables in order to be of interest to the scientific community in the field of exercise, since the current limitations mean that the results are of no particular relevance.

However, I leave it up to the editor to decide whether or not to publish it in this format and at this moment, according to the interest that the readers of IJERPH may have in this topic.

Reviewer 2 Report

The authors have addressed most of my comments within the restrains of their study design. This remains to be an unusual paper, in the sense that it does not present any data/results from a study per se, but rather describes aspects of the referral database. I will look forward to seeing more scientific output from this rich resource in the future.  

This manuscript is a resubmission of an earlier submission. The following is a list of the peer review reports and author responses from that submission.

Round 1

Reviewer 1 Report

I am grateful for the opportunity to review the paper entitled "The National ReferAll Database: An open dataset of exercise referral schemes across the UK". The authors have made a great effort to give visibility to the project that gives rise to the paper, with the aim of making the analysis of big data a useful tool for extensive use by the different professionals involved in physical exercise, as well as an instrument for visualizing the results obtained in the application of programs to increase physical activity.

Although it is a model that can provide relevant information to the field of study, there are several limitations to this design, which require clarification so that the data analyzed and obtained can be useful in the clinical practice of the professionals who wish to use it:

1.- The authors state "intervention fidelity is an issue of considerable importance for healthcare interventions. Considering this, a new framework for co-production of ERS alongside multidisciplinary stakeholders as a novel approach has been developed". Indeed, the lack of these data, or at least an approximation to them, may be a determining factor in obtaining some results or others, so perhaps it would be interesting to wait for them before publishing them. Is it possible to have these data currently available retrospectively? If so, it would be interesting to add them.

2- Intervention based on physical exercise does not present the same standardization when it is being applied as other types of treatment, so the results obtained must be adjusted to the type of exercise performed (type, duration, distribution, supervision, specific adjustment). Without these data, I do not consider that the results obtained can be extrapolated and, therefore, reproducibility is not possible.

3.- The paper has other limitations, as also indicated by the authors in limitations section, in reference to data on the subjects included, follow-up of the subjects (and reasons for their loss), ... The authors should consider the possibility of refining these limitations and presenting the results when it has been possible to incorporate these sections.

Improvements on these points would provide a very useful tool that, under these conditions, has a rather limited scope. The authors are encouraged to continue with the work, as it would be very useful if these limitations were solved.

Author Response

Response to all reviewers included in the attachment.

Reviewer 2 Report

Thank you for the opportunity to review this manuscript. I congratulate the authors on this well written and refreshingly transparent manuscript. The open approach to science taken throughout the manuscript, extensive efforts to make the dataset accessible and considerate suggestions have made this a particularly useful piece of work that will lay a good foundation for future analyses. Though part of me worries that the contents of the database will be misused by other researchers, I greatly appreciate the delicate wording regarding the limitations of the dataset and cautions about its use. I look forward to following the outputs from this database in the years to come.

There are no aspects of the manuscript that I feel must be changed prior to acceptance. 

One very minor suggestion is to review the index visits from the UK Biobank data that were used to derive the limits used for data cleaning for waist/hip circumference measures. I understand the largest dataset from UK Biobank was used though I suppose that the lower limits seem likely to be inch/cm conversion errors.

Author Response

Response to all reviewers available in the attachment.

Reviewer 3 Report

IJERPH-1111841 The National ReferAll Database: An open dataset of exercise referral schemes across the UK

This very interesting and unique paper that describes the formation and open sharing of The National ReferAll Database, data cleaning processes, and its structure including some outcome measures.

This is a rather unique type of paper as it mainly aims to describe what is available within the database, rather than do much analysis of the existing data.

Specific comments

Page 2: omit completely lines 84-89 and end your introduction with the actual aim of your study (as outlined on lines 79-80)

Table 1: it would be nice to be reminded which specific domain of health each one of these questionnaires assesses (add another column to table 1 with that info and include units for each variable).

Pages 12 and 13 and table 3: Are these baseline values? i.e. before the ERS’s took place? Or post interventions? Presented as they are in table 3 these data are almost meaningless. What would have been more interesting to see would be a comparison between pre and post intervention values adjusted for confounders? Otherwise, just a list of the data collected as outcome measures would suffice- I don’t see the need for the descriptive stats here.

Page 15: perhaps the most convincing outcome measure that would prove the efficacy of ERSs would be reduced GP visits, reduction in A+E visits etc- are these not captured by the database?

Page 17, lines 226-229- you are presenting new data (numbers) here- these belong in the results section and not in the discussion.

Page 17 Line 238 Does the database capture information about the types of exercise interventions implemented by different ERS? One of the major reasons for disappointing results on health and wellbeing outcomes could be that the interventions delivered are not appropriate. I agree that exercise in general improves health but that is such a generic statement and there is a lot of research to show that there is considerable variation in the efficacy of different types of exercise/ intensity/ frequency/ amount/ mode of delivery in relation to the outcome measure. For example, a falls prevention exercise programme would be completely different in its nature from a cardiac rehab exercise programme. Surely the national database is capturing that information. The above info would also be important if one was to carry out a cost effectiveness analysis of the schemes (which would be prudent).

Page 18 Line 280. Where in this manuscript have you presented pre-post comparison for the outcome measures?

Page 18, 1st vs 2nd paragraph- there is repetition of the same argument- issue of absence of control group

Author Response

(The authors gave the same response as above.)
